# CRISP - Complexity-based Reasoning of Internal Subprocessing

## Abstract

The growing deployment of *artificial intelligence* (AI) systems in safety-critical domains has underscored the need for transparent and trustworthy models. While existing explainability methods primarily focus on end-to-end interpretations, they often fall short of revealing the internal processing dynamics of deep networks. In this paper, we introduce CRISP a novel approach that decomposes neural networks into interpretable subprocesses, enabling a layer-wise analysis of hidden representations. Our method constructs interactive, low-complexity representations of input-output transformations within hidden layers, facilitating a deeper understanding of network behavior beyond final predictions. We present a framework and empirical validation for *Convolutional Neural Networks* (CNNs), demonstrating the method's potential to support more fine-grained, process-level insights into model operation.

## 1 Introduction

*Artificial Intelligence* (AI) systems are increasingly adopted in safety-critical domains such as autonomous driving (Ma et al., 2020; Levinson et al., 2011), medical diagnosis (Rajpurkar et al., 2022; De Fauw et al., 2018), industrial automation (Peres et al., 2020), and defense (Szabadföldi, 2021). In such contexts, the consequences of erroneous or non-transparent decisions can be severe, prompting regulatory and societal demands for explainability and trustworthiness (Doshi-Velez & Kim, 2017; Hine & Floridi, 2023; Comission, 2021; D'Elia, 2025).

To address these concerns, a broad range of *explainable AI* (xAI) and *Mechanistic Interpretability* (MI) techniques have been proposed. While these methods have output-level explanation, they generally lack mechanisms for analyzing the internal computations of a model (Zhang et al., 2021b;a; Zerilli et al., 2022). In particular, existing xAI methods are constrained by their reliance on output gradients or attribution signals, making them ill-suited for initiating analysis at hidden layers without a clear reference point. MI methods aim to address this but often suffer from limited generalization or ambiguous formalism (Sharkey et al., 2025).

We introduce **CRISP**, a visual analytics framework for *layer-wise functional decomposition* in deep neural networks. CRISP enables interpretability by identifying coherent subprocesses within hidden layers based on their computational characteristics. The idea of decomposing network behavior layer by layer is motivated by classical systems identification approaches in early computer vision (Haralick, 1992; Thacker et al., 2008; Greiffenhagen et al., 2001), where complex systems were analyzed through the modular breakdown of internal processes to infer component functionality. In parallel, the design of our approach is inspired by research in representation learning that demonstrates how probing local neighborhoods in feature space can reveal meaningful structure and geometry (Roweis & Saul, 2000; Tenenbaum et al., 2000; Belkin & Niyogi, 2003; Morcos et al., 2018; Raghu et al., 2017; Tishby & Zaslavsky, 2015).

At the core of CRISP is characterizing the local hidden-hidden activation space of individual hidden layers, enabling explainability to originate from within a layer. It is based on complexity analysis, using the $QI^2$ framework (Geerkens et al., 2024) on decomposed patches of the input and output feature maps. $QI^2$ quantifies the linearity of structural correlation between the input and output. It reveals latent structures and dependencies that represent spaces of computation that exhibit high internal coherence or strong deviations from the rest. Applied on layer-level, we aim at identifying the relevant patches that represent the layers transformation. Once identified, these spaces can be

used as layer-level entry point for downstream attribution or causal intervention. This facilitates the acquisition of a structural overview of the layer-wise computational behavior that bypasses the limitations of traditional end-to-end attribution pipelines.

In summary, we do not aim to provide another benchmarked attribution method. Instead, this paper introduces a new perspective on explainability: shifting the focus from global output explanations to localized, layer-level functional decomposition. CRISP provides a structured, layer-level entry point on top of which explanation methods or attributions can then be applied. To highlight the depth of insight our approach affords, we conduct exhaustive analyses of two networks rather than presenting a broad but shallow benchmark. We argue that this trade-off - depth over breadth - is essential for demonstrating the interpretive value of layer-level reasoning. Beyond these case studies, our framework opens up avenues for model debugging, compression, and modularity analysis at the level of internal transformations.

Our main contributions are as follows:

- We introduce a method for applying $QI^2$ to spatially decomposed input-output pairs at individual hidden layers, enabling direct characterization of their functional behavior.
- We demonstrate that $QI^2$ provides a generalizable descriptor for internal transformations, allowing conventional xAI methods to operate at hidden-layer level.
- We show that integrating $QI^2$ with existing explainability techniques enables fine-grained analysis of intermediate abstractions and processing steps across layers.
- We provide empirical evidence that deep neural networks can be decomposed into distinct, interpretable functional substructures at the hidden-layer level.

## 2 RELATED WORK

Early computer vision research went from benchmark-driven performance evaluation (Haralick, 1989; 1992) to modular performance evaluation, with pipelines that isolated stages such as feature extraction, matching, and classification (Greiffenhagen et al., 2001; Thacker et al., 2008). However, with the rise of deep learning, vision systems shifted from modular pipelines to holistic, end-to-end models (Krizhevsky et al., 2017). While this led to major performance gains, it obscured the roles of intermediate components. Initial evaluations remained output-focused, emphasizing predictive metrics without understanding internal computations.

Explainability research addressed the missing understanding in AI by introducing post-hoc attribution techniques. Gradient-based methods like *LRP* (Bach et al., 2015), and *CAM* (Zhou et al., 2016) traced the relevance from output to input. Perturbation-based approaches such as *LIME* (Ribeiro et al., 2016) and *SHAP* (Lundberg & Lee, 2017) used input modification to assess feature importance. However, these methods focus on output-driven explanations and do not offer explainability at the level of hidden-layer transformations.

A separate line of work uses deep visualization and feature inversion to study how internal representations relate to input space. *DeconvNet* (Zeiler & Fergus, 2014) projects intermediate feature maps back into pixels via a deconvolutional network, revealing which input structures activate a given unit or feature map without relying on class scores or final predictions. (Mahendran & Vedaldi, 2015) invert *deep representations* to analyze the information that is preserved across layers. Like our work, these methods aim to understand transformations along the forward pass but operate on the full activation space of a layer and do not provide a quantitative characterization of functionally coherent sub-transformations. Furthermore, concept-based methods such as *TCAV* (Kim et al., 2018) and *network dissection* (Bau et al., 2017) relate activations to human-interpretable concepts. Linear classifier probes (Alain & Bengio, 2018) examine the separability of representations at different depths employing small linear classifiers. *CRP* (Achtibat et al., 2023) extends LRP with layer-wise conditioning based on predefined concepts. Although approaching deeper analysis, these methods are limited by the need for human defined concepts or additional learning systems.

*Mechanistic interpretability* (MI) seeks internal decomposition via reverse engineering, investigating subnetworks and circuits (Olah, 2022; Olah et al., 2020). Approaches such as *Sparse Autoencoders* and *Dictionary Learning* extract monosemantic components from hidden activations (Huben et al., 2024; He et al., 2024; Bricken et al., 2023; Kissane et al., 2024), while *causal abstraction*

methods evaluate systemic behaviors through hypothetical interventions (Geiger et al.). *AttnLRP* adapts LRP to attention mechanisms in transformer architectures (Achtibat et al., 2024). These methods are promising, but typically architecture-specific and lack a shared formalism to define mechanistic explanations, limiting their generalization.

Despite progress in attribution, concept analysis, deep visualization and feature inversion, and MI, a general-purpose framework for interpreting *intermediate layer transformations* in a model-agnostic and functional way is still missing. CRISP bridges this gap by

1. enabling attribution grounded in the computational structure of hidden layers, independent of specific outputs or concept labels

2. providing complexity profiles that can guide deep visualization methods by quantitatively characterizing functionally coherent sub-transformations

Thus, CRISP provides a general approach to *layer-wise decomposition of AI functionality* while complementing existing visualization techniques.

## 3 PRELIMINARIES

We briefly summarize the $QI^2$ framework, which we use as the core complexity measure in CRISP. For details, we refer to (Geerkens et al., 2024).

**Global complexity score**. Let $P = \{p_1, \cdots, p_{|P|}\}$ be a set of data points, where each point is decomposed into input and output components $p_i = (v_{i1}, \cdots, v_{iI}, v_{o1}, \cdots, v_{oO})$ with input dimension I and output dimension O. We consider all ordered pairs $P^2 := \{x := (p_a, p_b) \mid p_a, p_b \in P\}$ and define pairwise distances in input and output space, $d_{RI}$ and $d_{RO}$, using task-appropriate metrics (e.g., Euclidean distance for vectors, SSIM for images). To make different data sets and input and output space comparable, distances are normalized by the mean distance over all pairs. For the input space,

$$d_{NRI}(x) := \frac{d_{RI}(x)}{\sum_{y \in P^2} d_{RI}(y)/|P^2|} \tag{1}$$

and analogously for the output space $d_{NRO}$. The global $QI^2$ score $QI^2R(P)$ measures the mismatch between relative input and output distances and thus the non-linearity of the input–output relationship defined as

$$QI^2R(P) := \frac{1}{|P^2|} \sum_{x \in P^2} \left(d_{NRI}(x) - d_{NRO}(x)\right)^2 \tag{2}$$

Low values indicate that relative input distances are well preserved in output space (locally "linear" behavior), while higher values indicate stronger non-linearities.

**Local complexity scores**. To obtain a spatially resolved view, $QI^2R(P)$ is evaluated locally over substructures. For each point $p_i \in P$ and every possible neighborhood size $k$, let $KNN(P, p_i, k)$ denote the subset containing of $p_i$ and its $k$ nearest neighbors (according to a chosen input space metric). The matrix of local $QI^2$ (**mlqi**$^2$) is defined as

$$\mathbf{mlqi}^2_{i,k}(P) := QI^2R(KNN(P, p_i, k)) \quad \forall p_i \in P, \ \forall k \in \{1, \ldots, |P| - 1\} \tag{3}$$

for all points indexed with $i$ and neighborhood sizes $k$. The $MLQI^2$ thus provides a local complexity profile over the data, quantifying how (non-)linear the input-output relationship is within a local neighborhood around $p_i$. For visualization, these local values are aggregated into a histogram over complexity bins $v$ (y-axis) and neighborhood sizes $k$ (x-axis). Let $\mathbf{hlqi}^2(P)$ denote the deduplicated counts of local complexities $MLQI^2$ falling into bin $v$, the normalized, gamma-corrected histogram is defined as

$$\mathbf{shlqi}^2_{v,k}(P) := \left( \frac{\mathbf{hlqi}^2_{i,k}(P)}{\sum_{s=1}^{|P|} \mathbf{hlqi}^2_{s,k}(P)} \right)^{\gamma} \tag{4}$$

## 4 METHODOLOGY

We consider a neural network as a high-dimensional function $\mathcal{F}_\theta : \rho(\mathcal{I}) \to \rho(\mathcal{O})$, mapping an input $\mathbf{I} \in \mathcal{I}$ to an output $\mathbf{O} \in \mathcal{O}$. The function $\mathcal{F}_\theta$ is composed of $L$ successive (or parallel) transformations, such that $\mathcal{F}_\theta = f_{\theta_L}^L \circ f_{\theta_{L-1}}^{L-1} \circ \ldots \circ f_{\theta_1}^1$, where each $f_{\theta_l}^l : \mathbf{z}^{l-1} \to \mathbf{z}^l$ represents an individual layer or computational unit parameterized by $\theta_l$ mapping input $\mathbf{z}^{l-1}$ to output $\mathbf{z}^l$. Motivated by principles from system identification in classical computer vision pipelines, we conceptualize each $f_{\theta_l}^l$ or groups of adjacent layers as functional *module*.

For a given *module*, we propose a three-stage process illustrated in fig. 1. The key idea is that the functional behavior of a *module* is reflected in the structure of its local input-output transformations. Prior work in manifold learning and deep representation analysis has demonstrated the interpretive value of probing local neighborhoods in feature space (Roweis & Saul, 2000; Tenenbaum et al., 2000; Belkin & Niyogi, 2003; Morcos et al., 2018; Raghu et al., 2017; Tishby & Zaslavsky, 2015). First, during inference, we extract and decompose the input and output activations of a target *module* into patches (**Decomposition of input and output**). These decomposed patches form the basis for a complexity analysis using QI$^2$, which is used to select patches based on their transformation characteristics (**Complexity representation**). Finally, we employ these selected patches as anchors for backward attribution and interpretation (**Function reasoning**). This process can be seen similarly to the standard procedures of xAI methods, in which either the final prediction or intermediate feature spaces are chosen as anchor for explanations. QI$^2$ serves as focused entry point for this anchor at the layer-level.

**Decomposition of input and output.** To analyze the function of a given *module*, we first decompose its input $\mathbf{z}^{l-1}$ and output $\mathbf{z}^l$ into corresponding spatial or semantic *queries* $\mathbf{q}$. We define the resulting decomposition as the set

$$\mathcal{D} := \left\{ \mathbf{q}_p \mid p \in \mathcal{P}, \mathbf{q}_p := \begin{bmatrix} \mathbf{z}_\mathbf{p}^{l-1} \\ \mathbf{z}_\mathbf{p}^l \end{bmatrix} \right\} \tag{5}$$

where $\mathbf{z}_p^{l-1}$ denotes the flattened $p$-th localized patch from the input, and $\mathbf{z}_p^l$ the corresponding flattened patch from the output. Crucially, each input is paired with its corresponding output *as defined by the computation of the* module *itself* (e.g., spatial tiling with convolutional receptive fields in CNNs, attention-weighted tokens in transformers). Thus, every query reflects the actual local transformation carried out by the *module*, ensuring that the decomposition is fully consistent with the underlying operation. The index set $\mathcal{P}$ defines all valid positions or units for which paired patches can be extracted.

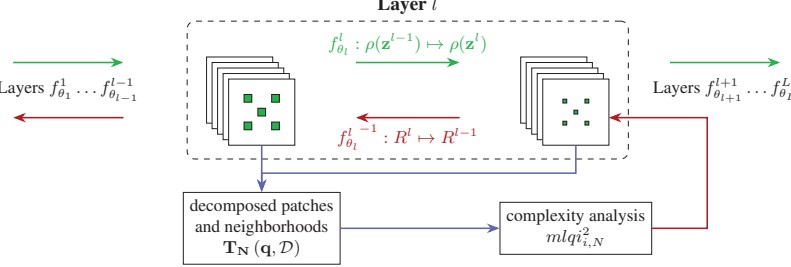

Figure 1: Schematic illustration of the proposed methodology for layer-wise process analysis using QI$^2$. Green arrows represent forward inference; blue arrows denote **decomposition of input and output** and **complexity representation**; red arrows indicate **function reasoning** via backward attribution.

In this work, we apply our approach to convolutional layers, with full details and examples for decomposition provided in the appendix.

**Complexity representation.** For each query $\mathbf{q} \in \mathcal{D}$, we construct multiple individual local tuples of $N$ nearest neighbors:

$$\mathbf{T_N}\left(\mathbf{q}, \mathcal{D}\right) = \left\{ \mathbf{q}_{c_i} \mid c_i \in \{c_0, c_1, \ldots, c_N\} \right\}, \tag{6}$$

determined by the input distance metric $d_{NRI}\left(.,.\right)$ (here: Euclidean distance). Each tuple $\mathbf{T_N}$ forms a local subspace of transformations for which we compute a complexity value $\mathrm{QI}^2\mathrm{R}\left(\mathbf{T_N}\right)$, which quantifies how (non-)linear the relationship is between input and output space within the subspace. This systematic evaluation obtains the $\mathrm{MLQI}^2$

$$\mathbf{mlqi}_{i,N}^2\left(\mathcal{D}\right) \coloneqq QI^2R\big(\mathbf{T_N}\left(\mathbf{q}_i, \mathcal{D}\right)\big) \quad \forall \mathbf{q}_i \in \mathcal{D},\ \forall N \in \{1, \ldots, |\mathcal{D}| - 1\} \tag{7}$$

where $i$ indexes the query $\mathbf{q}$ and $N$ the number of neighbors considered in the tuple $\mathbf{T_N}$.

The $\mathrm{MLQI}^2$ comprises a locally resolved complexity profile of the input-output transformation of the *module*, providing one complexity value for each subspace spanned by $N$ neighbors around each query point $\mathbf{q}_i$. This delineates the degree to which the *module*'s operation exhibits (non-)linearity in comparison to the adjacent $N$ inputs.

To visually analyze the local complexity profile, we use the interactive three-dimensional histogram $\mathrm{SHLQI}^2$ (Geerkens et al., 2024). The resulting $\mathrm{SHLQI}^2$ provides a compact complexity profile of the *module* that includes interesting patterns like dark, coherent ridges that denote coherence and persistence of neighborhoods that share similar transformation behavior, isolated peaks indicating outlier-like transformations, or deviations from the overall distribution that highlight specialized local processes.

Importantly, both $\mathrm{MLQI}^2$ and $\mathrm{SHLQI}^2$ retain direct correspondence to the queries. Each complexity value can be traced back to its originating subspace $\mathbf{T_N}$ and associated query.

**Function reasoning.** The identified patterns of $\mathrm{SHLQI}^2$ are leveraged to uncover the specific functional behavior encoded by tracing back the queries that give rise to those patterns. In particular, these salient regions of $\mathrm{SHLQI}^2$ serve as anchors for explainability approaches depending on the architecture and operation of the *module* (e.g., LRP, TCAV, DeconvNet).

**Workflow.** CRISP is applied to an entire architecture through the following practical workflow:

1. **Layer-level scan:** For a given architecture, we compute $\mathrm{SHLQI}^2$ for all layers of a chosen type (e.g., convolutional layers, encoder/decoder blocks) over a representative input set. This yields a compact "complexity profile" of the network and an initial processing chain.

2. **Layer selection:** We identify layers with pronounced hotspots or distinctive coherent processing characteristics and focus detailed analysis on them.

3. **Subprocess extraction:** Within those layers, we conduct an exhaustive analysis over a multitude of inputs to strengthen hypothesis of processing.

4. **Linking across layers:** We then connect these subprocesses along the network to form processing chains

**Summary:** The computation and investigation of the $\mathrm{SHLQI}^2$ histogram serves as entry point of explainability tools. By aggregating and comparing the results in multiple input samples, we form a descriptive explanation of the computational role of each layer. This offers a new explainability pathway: rather than asking *what output is caused by this input*, we instead ask *what part of the input triggers this internal function*, as revealed through localized complexity and backward attribution.

For this paper, we perform backward attribution using Layer-wise Relevance Propagation (LRP), adapted to work on hidden layers within the network. Specifically, we employ the $\mathrm{LRP}_{\mathrm{CMP}}$ gradient attribution, as suggested by the developers in their best practices paper Kohlbrenner et al. (2020). The implementation is based on a customized version of the `Captum` (v0.7.0) Kokhlikyan et al. (2020) library. This attribution yields a heat map over the input, highlighting which input regions contributed most to the triggering of a transformation behavior represented by the identified patches.

## 5 EXPERIMENTS

We demonstrate the explanatory power of CRISP through two representative case studies using convolutional neural networks (CNNs) of differing complexity. First, we analyze a VGG16 architecture trained for age classification, replicating the setup in Lapuschkin et al. (2017). This allows us to contrast our functional decomposition approach with standard explanation methods such as (LRP).

Second, we apply our methodology to a more complex U-Net architecture used for semantic railway segmentation. This architecture enables a detailed inspection of how different submodules contribute to structured outputs, and reveals functionally significant components.

In each case, we apply our analysis to a broad set of inputs for every module, and the visualizations shown are selected as representative examples from this larger pool. For a given module, the same complexity hotspots and subprocess patterns reappear consistently across heat maps, and we only report qualitative patterns once we have verified that they are stable across many inputs to describe the functional behavior. Thus, the examples in should be interpreted as canonical views of the module's behavior rather than cherry-picked outliers.

### 5.1 AGE AND GENDER CLASSIFICATION VGG16

In our first case study, we analyze a VGG16 network for age classification. This experiment serves to illustrate how our approach provides deeper insights than standard xAI methods, specifically comparing the functional breakdown provided in this chapter against the LRP-based analysis in Lapuschkin et al. (2017). Since the original implementation was not publicly available and used a different deep learning framework, we retrained the VGG16 model from scratch using PyTorch, matching the hyperparameters reported in the original study.

**Early Layers: Skin Tone and Face Region Detection.** In the first few convolutional layers, the network predominantly learns to detect regions of uniform skin tone, both brighter and darker, as a proxy for localizing the face. This is illustrated in Figure fig. 2, where a face lit unevenly by sunlight yields distinct activations. The $SHLQI^2$ histograms in this figure correspond to the same convolutional layer, with different regions marked for analysis. The corresponding heatmaps reveal input attributions that highlight these skin-tone-based areas. Across the first three convolutional layers, this functionality remains stable and can be interpreted as a low-level face detection mechanism based on color homogeneity.

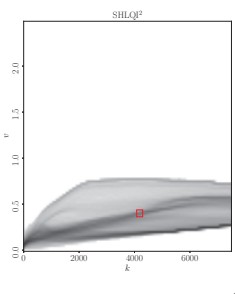 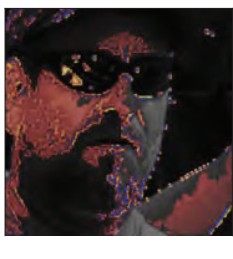     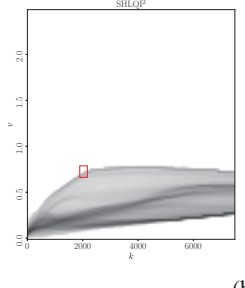 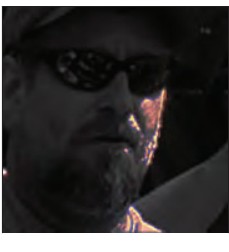

(a)                                    (b)

Figure 2: $SHLQI^2$ and heat maps regarding the marked areas for the first convolution layer. (a) shows $SHLQI^2$ and heat map for the marked middle area, (b) shows $SHLQI^2$ and heat map for the marked top area

**Mid-Level Layers: Facial Feature Extraction.** The subsequent seven convolution layers focus on detecting contrastive edges that correspond to facial features such as eyes, nose, and mouth. Figure fig. 3 shows one representative example, where both the $SHLQI^2$ histogram and the heatmap indicate strong selectivity for edge-like structures. This reflects a mid-level functional decomposition where facial components are separated based on local texture contrast and structure.

**Late Layers: Feature Emphasis and Class-Specific Processing.** In the final convolutional stages, the network appears to combine and emphasize previously detected features, particularly those lo-

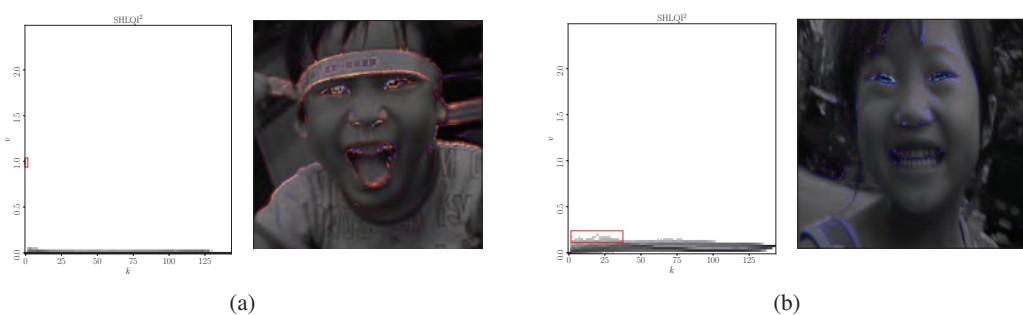

(a)                                                                              (b)

Figure 4: SHLQI$^2$ and heat maps regarding the marked areas for the tenth convolution layer (a) and the eleventh convolution layer (b).

cated within the face region. Figure fig. 4 illustrates this behavior: the heatmaps show enhanced relevance within facial areas, although the contrast between relevant and irrelevant regions is not strongly pronounced. This limited focus may indicate a bottleneck in the model's capacity to emphasize semantically meaningful features for the classifier, possibly contributing to its suboptimal performance.

Compared to the single-image explanation offered in Lapuschkin et al. (2017) figure 5, our results presented in this chapter not only reproduce the general findings, but also offer a more fine-grained functional breakdown than a single output-dependent heatmap. By combining SHLQI$^2$ histograms with backward attributions, we can trace the emergence of specific behaviors across layers. This enables a layer-wise, interpretable explanation that aligns local complexity with functional roles, providing clearer insights into the model's internals.

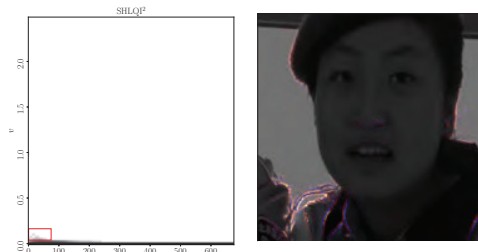

Figure 3: SHLQI$^2$ and heat map regarding the marked area for the seventh convolution layer.

## 5.2 RAILWAY SEGMENTATION BASED ON U-NET

In a more complex experiment, we applied our analysis to a U-Net architecture designed for railway segmentation in images trained on the Railsem19 dataset (Zendel et al., 2019). We examined all convolutional layers, as well as all upsampling and downsampling blocks, across approximately 30 images from Railsem19 and OSDaR23 (Tilly et al., 2023).

To contextualize our findings, we briefly outline the architecture of the employed U-Net. A full visualization is provided in the appendix. The network begins with an initial downsampling module (`Down`), followed by a convolutional module (`Inc`) that maintains spatial dimensions. This is followed by four downsampling blocks (`Down1` to `Down4`), each consisting of max-pooling layers for spatial reduction and convolutional, normalization, and activation layers for feature extraction. Feature map dimensions are halved at each pooling stage. The decoder mirrors this structure with four upsampling blocks (`Up1` to `Up4`), where each block includes upsampling, convolution, normalization, and activation layers, doubling spatial dimensions accordingly. For clarity, we refer to individual layers by combining their module name, type, and index (e.g., `Down_Conv1`). Where entire modules are discussed, we refer to them by name only (e.g., `Up2`).

Our analysis revealed four distinct stages in the network's processing pipeline, each corresponding to a different aspect of the segmentation task.

The first processing stage includes the initial modules (`Down`, `Inc`, `Down1`, and `Down2`). These layers primarily perform sky detection. Depending on the input image, three characteristic processing patterns were identified:

- **Homogeneous skies**: At the SHLQI$^2$ level all modules in this group exhibit steep increases in value followed by abrupt complexity drops, suggesting a classification-like internal dichotomy (fig. 5a, fig. 5b). At the functional level, these layers separate sky-related pixels from the rest, consistently assigning sky patches to a distinct regime used later to suppress rail candidates in the sky region

- **Skies with small gradients**: Initial layers behave similarly, but subsequent modules show a weaker rise in complexity, indicating a transition toward a regression-like process.

- **Skies with strong gradients**: Here, early modules do not clearly exhibit sky detection behavior. Sky-related features are instead only present in later modules.

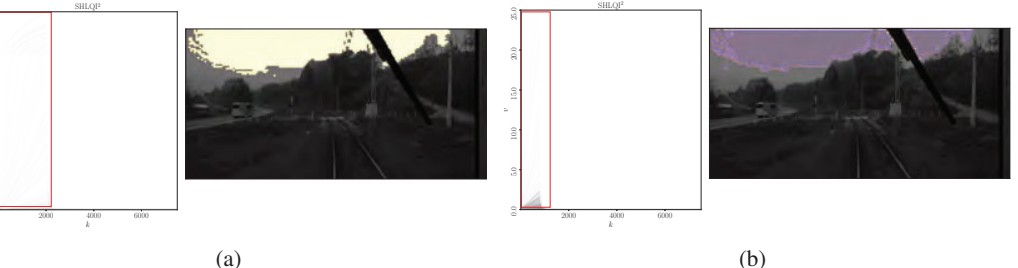

(a)          (b)

Figure 5: SHLQI$^2$ and respective heat map for marked area of layers `Down_Conv0` (a) and `Down1_Conv0` (b).

Rail-specific features begin to emerge starting with module `Down3_Conv0` and continue through `Down4_Conv1`. In fig. 6a, for instance, the SHLQI$^2$ shows gradually increasing complexity in areas where distinct lines intersect with orthogonal structures suggesting the detection of start or termination of rails. The corresponding heat map confirms this behavior.

In `Down4`, the SHLQI$^2$ curves display localized complexity peaks at low neighborhood sizes, which correspond to features such as double-gradient edges (e.g., bright–dark–bright or dark–bright–dark), typical of rail surfaces. Sky-related features are largely absent at this stage, reflecting a shift in processing focus from context to object-level structure, as can be seen in fig. 6b.

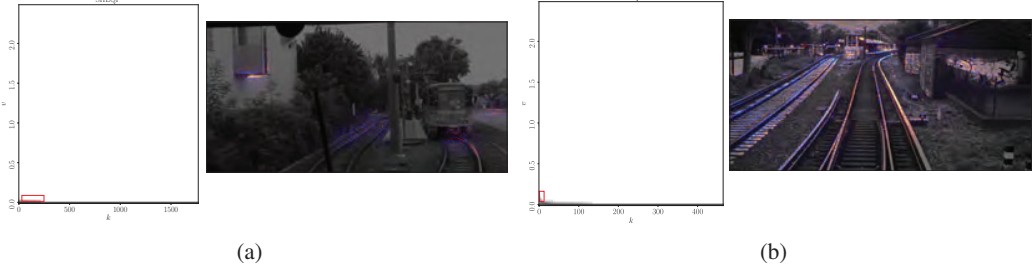

(a)          (b)

Figure 6: SHLQI$^2$ and respective heat map for marked area of layers `Down3_Conv0` (a) and `Down4_Conv1` (b).

Modules `Up1` and `Up2` reconstruct a coherent railway region using the previously extracted characteristics. The detected rail endpoints and double-gradient edges are aggregated into complete rail structures. Simultaneously, the network identifies surrounding context, such as gravel, grass, or sleepers, to delineate railway regions. This process is visible in heat maps such as fig. 7a, where emphasis is placed on regions adjacent to the rails. Sky information is still partially retained due to skip-connections, but is gradually suppressed in favor of railway-specific features, as seen in fig. 7b.

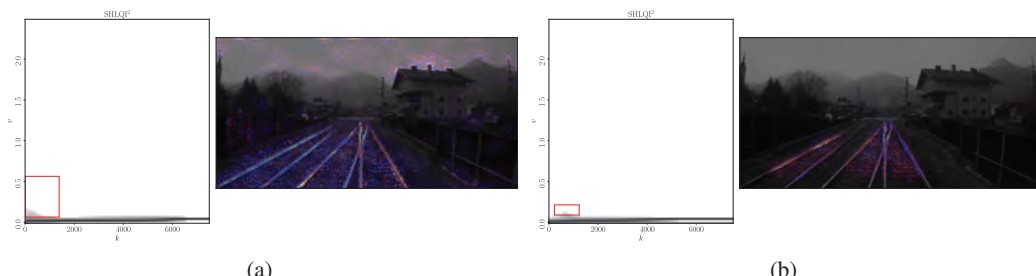

(a)                                                        (b)

Figure 7: SHLQI$^2$ and respective heat map for marked area of layers `Up2_Conv0` (a) and `Up2_Conv1` (b).

To demonstrate how skip-connections contribute to this process, we analyzed both inputs of `Up2` independently. The output from the previous decoder stage (input 0) focuses on constructing rail structures as can be seen in fig. 8a. Meanwhile, the skip-connected input (input 1) retains earlier sky detection, although with increasing suppression of sky regions, illustrating the interplay between context and refinement in this phase (fig. 8b).

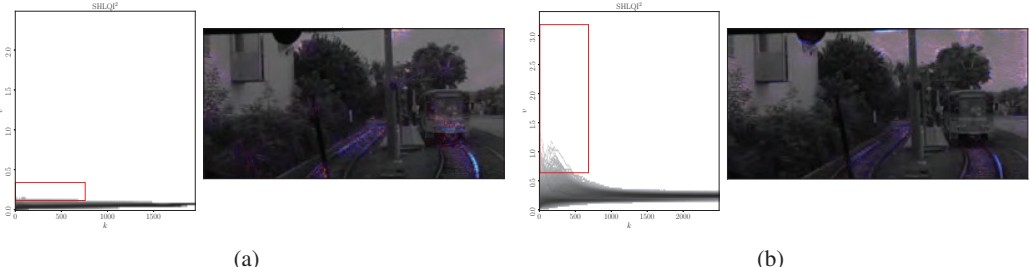

(a)                                                        (b)

Figure 8: SHLQI$^2$ and respective heat map for marked area for first input (from skip connection) (a) and second input (from previous layer) (b) of layer `Up2`

The final stage, carried out in `Up3` and `Up4`, refines the segmentation output to accurately highlight the railway area located between parallel rails. In fig. 9, the heat map clearly shows that processing is now focused almost exclusively on this narrow strip. At this point, earlier context information has been fully filtered, and the model concentrates on producing pixel-level precision within the designated railway corridor.

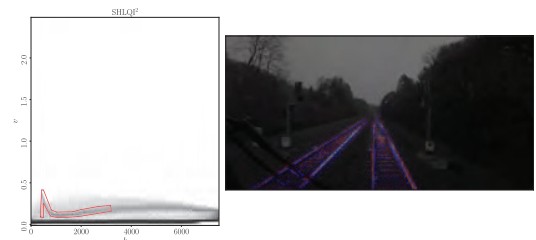

Figure 9: SHLQI$^2$ and respective heat map for marked area of layer `up3_Conv1`.

The U-Net follows a systematic processing strategy for railway segmentation:

- **Sky detection** is performed early to aid in later region filtering.

- **Initial rail detection** identifies boundaries using structural features such as orthogonal lines and double-contrast edges.

- **Railway region construction** combines detected rails with contextual cues to highlight the railway environment.

- **Final refinement** isolates the railway corridor with high spatial accuracy.

This structured sequence of processes demonstrates the network's capacity for both contextual understanding and spatial precision, aligning with U-Net's strengths in semantic segmentation tasks.

## 6 DISCUSSION

In this work, we introduced CRISP for the analysis of hidden-layer transformations based on complexity representations. Using $QI^2$ to quantify the structural properties of input-output transformations at the layer-level, our approach enables a new form of explainability beyond end-to-end explanations. We exemplary used CRISP on CNNs in combination with LRP, demonstrating its feasibility.

Our key findings include:

- A representation using $QI^2$ serves as a meaningful descriptor of transformation within individual hidden-layers, allowing for the identification of coherent computational units.
- Integration of $QI^2$ with established attribution methods such as LRP enables layer-wise input attribution, extending the scope of use of traditional explainability techniques.
- The resulting explainability does not rely on handcrafted concepts, predefined labels, or additional learning systems, offering a more intrinsic approach to xAI.

Unlike attribution-based xAI methods, CRISP does not primarily aim to quantify which inputs influence outputs. Instead, it seeks to uncover the functional roles of internal modules. This makes standard quantitative benchmarks (e.g., pixel attribution fidelity scores) less appropriate. They would first need to be reframed in terms of (i) coherence and stability of local transformation regimes and (ii) their alignment with known internal structure. For the current submission, quantitative measures would not post a benefit in understanding the approach or presentation of the findings. We see CRISP as a complement, not a competitor, to attribution-based evaluation.

**Potential Applications.** CRISP has natural applicability in domains where internal network verification is critical. In medical imaging, for instance, CRISP could be used to examine whether early layers of a diagnostic model consistently detect anatomical boundaries before higher-level inference is performed—supporting regulatory auditing and expert validation. In autonomous driving, identifying whether modules consistently capture safety-critical features such as road edges, lane markings, or dynamic objects across layers could contribute to transparency and safety certification. The ability to isolate and interpret individual processing steps opens the door for CRISP to become a practical tool in trustworthy AI pipelines, extending its utility beyond academic settings.

**Limitations and Future Work.** The computational demands for computing $QI^2$ across several hidden layers with a currently not optimized implementation is substantial, both in terms of runtime and hardware requirements. For now, scalability remains a concern for very deep or high-resolution models that needs to be improved as future work.

While the current implementation involves manual inspection of $SHLQI^2$ histograms and heatmaps, this process is conceptually similar to other xAI methods such as LRP or Grad-CAM, which also require manual selection of starting points and visual interpretation of heatmaps. CRISP differs mainly in the scope of repeated inspection across multiple layers and samples. Importantly, the structured nature of $SHLQI^2$ suggests promising avenues for automating the selection process by optimizing objectives over $QI^2$ and thus bypass explicit visual inspection for downstream tasks, which we leave for future work.

Although our experiments focus on CNNs, CRISP is in *principle* not architecture-specific: any model with modular hidden transformations (e.g., self-attention in transformers, message passing in GNNs, recurrent transitions) can, in principle, be analyzed. From a practical standpoint, extending CRISP to additional architectures mainly requires engineering effort in defining suitable "patches" for their hidden representations, integrating with their attribution tools, and running large-scale experiments. Demonstrating this generality empirically remains important future work.

In conclusion, we believe that complexity-based representations such as $QI^2$ can serve as a unifying abstraction for the analysis of hidden-layer behavior. By extending the frontier of explainability to the internal mechanisms of AI systems, our work contributes a valuable building block toward truly transparent and trustworthy AI.

## ETHICS STATEMENT

This work aligns with the principles outlined in the ICLR Code of Ethics. By focusing on interpretability at the layer-level, our method contributes to responsible AI research that prioritizes transparency, trustworthiness, and accountability. Enhanced interpretability supports the goal of ensuring AI systems benefit society while minimizing potential harm, particularly in safety-critical domains such as healthcare or autonomous driving. We have upheld standards of scientific integrity by accurately reporting methods and results, and we acknowledge the importance of fairness, inclusivity, and respect for data privacy in future applications of this work.

This work makes use of the publicly available Adience dataset for age and gender classification. The dataset contains unblurred images of faces collected from Flickr albums, and was released under terms permitting research use. We acknowledge the potential privacy implications of using identifiable facial data and note that our experiments were conducted strictly within the scope of academic research. No attempt was made to identify or re-associate individuals, and results are reported only in aggregate. We further emphasize that any downstream use of our method in real-world contexts must consider ethical safeguards, particularly with respect to fairness and potential societal impacts of automated age and gender recognition systems.

## REPRODUCIBILITY STATEMENT

To reproduce the results from this paper, we provide a standalone part of our software, which can be downloaded anonymously here (just click on the name `CRISP_ICLR` at the top and choose download). It contains the software with which we conducted the analysis of the U-Net railway segmentation model. We include the software for necessary computations that are mentioned in the paper along with a few small exemplary files for the analysis of the U-Net. Further information on how to use this software are given in the appendix and the `README.md` file within the software.

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

APPENDIX

USE OF LLMS

In this paper, the use of LLMs was minimal in the processes of compressing and adapting the style of writing single sentences or paragraphs. There was no significant contribution at the level of a contributing author.

DECOMPOSITION OF INPUT AND OUTPUT FOR CNNS

Restricted to CNNs, we have the operational constraint of two-dimensional inputs $(h, w)$ to either the network itself or the module, with a third dimension depth $c$. For each $f_l$, we can spatially decompose $\mathbf{z}^{l-1}$ and $\mathbf{z}^l$ similarly to the N-Tuple subspace classifier Bledsoe & Browning (1959) and use the decomposed patches to investigate the feature encoded by the module. We define the set of patches

$$\mathcal{D} = \left\{ (\mathbf{z}_p^{l-1}, \mathbf{z}_p^l) \mid p \in \mathcal{P} \right\} \tag{8}$$

where $\mathbf{z}_p^{l-1}$ denotes a patch from the input of the module and $\mathbf{z}_p^l$ denotes the corresponding patch from the output of the module. $\mathcal{P}$ is a set of valid indices for which patches can be extracted from both input and output, depending on the patch size and stride of the module. To exactly represent the transformation of the module, we need to specify the spatial position in the input and output and the spatial dimensions of the patches concretely.

For this, we define

- the aggregated receptive field in the input ($\mathbf{r}^{l-1}$), which represents the spatial region in the original input that contributes to the activation of a particular output in $\mathbf{z}^l$
- the activated field in the output ($\mathbf{r}^l$), which represents the spatial extent within $\mathbf{z}^l$ that is directly influenced by a particular input in $\mathbf{z}^{l-1}$
- the input stride ($\mathbf{s}^{l-1}$)
- the output stride ($\mathbf{s}^l$)

A pseudocode for computing these values is given in algorithm 1.

---

**Algorithm 1** Computations of spatial sizes and positions

---

**for all** Input $\mathbf{z}^{l-1}$ **do**
    $\mathbf{r}^{l-1} \leftarrow 1, \mathbf{r}^l \leftarrow 1, \mathbf{s}^{l-1} \leftarrow 1, \mathbf{s}^l \leftarrow 1$
**end for**
**for all** modules $mod$ during forward **do**
    Extract kernel size $k$, stride $s$, scale factor $sc$
    **if** $mod$ is Container **then** continue
    **else if** $mod$ is Convolution or $mod$ is Pooling **then**
        $\mathbf{r}^{l-1} \leftarrow \mathbf{r}^{l-1} + (k - 1) \cdot \mathbf{s}^{l-1}$
        $\mathbf{s}^{l-1} \leftarrow \mathbf{s}^{l-1} \cdot s$
    **else if** $mod$ is TransposedConvolution **then**
        $\mathbf{r}^l \leftarrow \mathbf{r}^l + (\mathbf{r}^l - 1) \cdot s + (k - 1) + 1$
        $\mathbf{s}^l \leftarrow \mathbf{s}^l \cdot s$
    **else if** $mod$ is Upsample **then**
        $\mathbf{r}^l \leftarrow \mathbf{r}^l \cdot sc$
        $\mathbf{s}^l \leftarrow \mathbf{s}^l \cdot sc$
    **end if**
**end for**

---

Given the defined strides in input $\mathbf{s}^{l-1}$ and output $\mathbf{s}^l$, the receptive field $\mathbf{r}^{l-1}$ and the activated field $\mathbf{r}^l$, we can concretely define the spatial locations of the patches in the input and output.

$$\mathcal{P} := \left\{ (i,j) \in \mathbb{N}_0^2 \mid 0 \le i \le \widehat{h}, \ 0 \le j \le \widehat{w} \right\} \tag{9}$$

Where $\widehat{h} = \left\lfloor \frac{h-\mathbf{r}}{\mathbf{s}} \right\rfloor$ and $\widehat{w} = \left\lfloor \frac{w-\mathbf{r}}{\mathbf{s}} \right\rfloor$ denote the number of patches possible in spatial dimensions height and width. Finally, the spatially decomposed patches of $\mathbf{z}^{l-1}$ and $\mathbf{z}^l$ can be expressed as:

$$\mathcal{D} = \left\{ \left( \mathbf{z}_{i \cdot \mathbf{s}^{l-1}, j \cdot \mathbf{s}^{l-1}}^{l-1}, \ \mathbf{z}_{i' \cdot \mathbf{s}^l, j' \cdot \mathbf{s}^l}^l \right) \mid \right.$$
$$\left. (i,j) \in \mathcal{P}^{l-1}, \ (i',j') \in \mathcal{P}^l \right\} \tag{10}$$

where $\mathbf{z}_{i \cdot \mathbf{s}_1^{l-1}, j \cdot \mathbf{s}_2^{l-1}}^{l-1}$ and $\mathbf{z}_{i' \cdot \mathbf{s}_1^l, j' \cdot \mathbf{s}_2^l}^l$ denote the spatially decomposed input and output patches at spatial indices $(i,j) \in \mathcal{P}^{l-1}$ for the input and $(i',j') \in \mathcal{P}^l$ for the output, retaining their depth and local receptive field size $\mathbf{r}^{l-1}$ and activated field size $\mathbf{r}^l$.

### RAILWAY SEGMENTATION NETWORK

The aforementioned analyzed network for railway segmentation is based on a modified U-Net architecture, with add-on layers upfront and at the end. fig. 10 shows a brief visualization of the real structure. The network was trained on the RailSem19 Zendel et al. (2019) dataset. We unfortunately cannot give further information about hyperparameter selection or other technical details of training and evaluation due to confidentiality constraints.

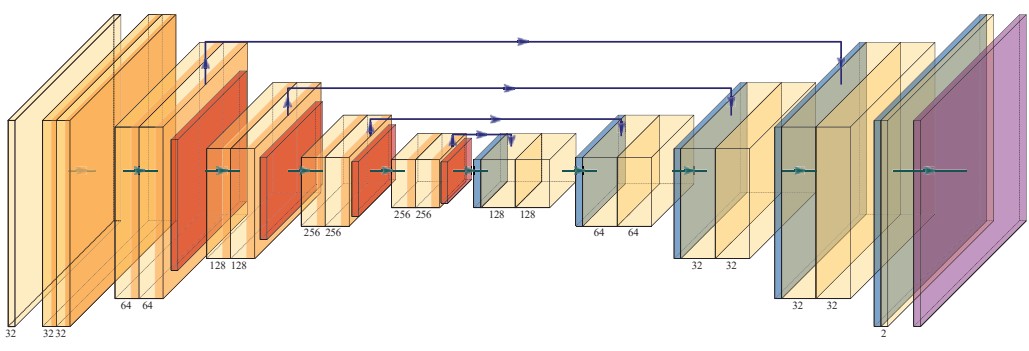

Figure 10: Railway segmentation U-net architecture

### SOFTWARE AND REPRODUCIBILITY

For proper use of the software, a `README.md` file is included, which covers the installation and use of different parts of the software for computation and analysis.

To reproduce some results of this paper, we included a few files within the software that represent our analyzes. The files include the data and the $SHLQI^2$ presented in the paper but in a reduced manner. Due to the file size limit for the supplementary, we only considered up to 1000 neighbors during $SHLQI^2$ computation, in contrast to 7500 in the paper. For analysis of previously computed data and $SHLQI^2$, one has to go to the top level folder of the software provided and start the script `UNet_nn_analysis.py` as state in the `README.md`. The arguments `IMAGE`, `LAYER`, `INPUT`, `stride_x`, and `stride_y` determine the functionality.

- `IMAGE`: the image used as baseline for computation. Available are:
  - `OSDAR_017_0924`
  - `rs00004.jpg`

- `rs00018.jpg`
- `rs00420.jpg`
- `rs00506.jpg`

- `LAYER`: the layer to be analyzed. Available are:
  - `down1-maxpool_conv-1-conv-Conv2d0`
  - `down3-maxpool_conv-1-conv-Conv2d0`
  - `down4-maxpool_conv-1-conv-Conv2d0`
  - `Up0`
  - `up2-conv-conv-Conv2d0`
  - `up3-conv-conv-Conv2d1`

  Default is `down1-maxpool_conv-1-conv-Conv2d0`

- `INPUT`: the index of the input to the network (change only for `Up` modules)

- `stride_x`, `stride_y`: strides multipliers in x and y direction. Given in table 1

After a few seconds, two windows will appear. The first contains the visualization of the data and the $\text{SHLQI}^2$ histogram, the second contains the space for the generated heatmap. By clicking and dragging the mouse over the histogram of $\text{SHLQI}^2$, you can now choose an area, similar to the red boxes within the paper, to be attributed back to the input image. The attributed heat map will be shown in the other window.

To compute the data for the entire network, the script `UNet_nn_computations.py` is provided. With this script, the software computes data and $\text{SHLQI}^2$ for every layer of the given type for all images within the folder `\CRISP\tests\UNet\images` and stores the files into a new folder under `\CRISP\tests\UNet\QI2_files\<imagename>`.

| layer name | stride_x | stride_y |
|---|---|---|
| down-Conv2d0 | 3 | 3 |
| inc-conv-Conv2d0 | 3 | 3 |
| inc-conv-Conv2d1 | 3 | 3 |
| inc-conv-Conv2d2 | 3 | 3 |
| inc-conv-Conv2d3 | 3 | 3 |
| down1-maxpool_conv-1-conv-Conv2d0 | 2 | 2 |
| down1-maxpool_conv-1-conv-Conv2d1 | 2 | 2 |
| down2-maxpool_conv-1-conv-Conv2d0 | 1 | 1 |
| down2-maxpool_conv-1-conv-Conv2d1 | 1 | 1 |
| down3-maxpool_conv-1-conv-Conv2d0 | 1 | 1 |
| down3-maxpool_conv-1-conv-Conv2d1 | 1 | 1 |
| down4-maxpool_conv-1-conv-Conv2d0 | 1 | 1 |
| down4-maxpool_conv-1-conv-Conv2d1 | 1 | 1 |
| up1-conv-conv-Conv2d0 | 1 | 1 |
| up1-conv-conv-Conv2d1 | 1 | 1 |
| up2-conv-conv-Conv2d0 | 1 | 1 |
| up2-conv-conv-Conv2d1 | 1 | 1 |
| up3-conv-conv-Conv2d0 | 2 | 2 |
| up3-conv-conv-Conv2d1 | 2 | 2 |
| up4-conv-conv-Conv2d0 | 3 | 3 |
| up4-conv-conv-Conv2d1 | 3 | 3 |
| outc-Conv2d0 | 5 | 5 |
| Up0 | 1 | 1 |
| Up1 | 1 | 1 |
| Up2 | 2 | 2 |
| Up3 | 3 | 3 |
| Down0 | 2 | 2 |
| Down1 | 1 | 1 |
| Down2 | 1 | 1 |
| Down3 | 1 | 1 |

Table 1: Layer names and default strides used for the analysis in the main paper

