# OpenReview forum: "CRISP - Complexity-based Reasoning of Internal Subprocessing"
_ICLR.cc/2026/Conference — Submitted to ICLR 2026_

### Official Review · Reviewer_jTky · 2025-10-28

**Soundness:** 3
**Presentation:** 1
**Contribution:** 2
**Rating:** 2
**Confidence:** 4

**Summary:**

The paper proposes an XAI method for reasoning at the intermediate-layer level. The authors leverage the $QI^2$ metric, which quantifies the nonlinearity between the input and output spaces based on local input-output transformations, to construct anchors for XAI methods (feature attribution is performed using LRP).

The approach is demonstrated on multiple examples using VGG-16 (trained for age classification) and U-Net (trained for railway segmentation). The results indicate that the proposed method successfully identifies meaningful and informative features.

**Strengths:**

1. The paper sheds light on explanations in hidden layers - an interesting and emerging subdomain within the broader field of XAI.
2. The qualitative analysis (right side of Figures 2-7) provides interesting insights and demonstrates the potential of the proposed method.

**Weaknesses:**

1. Contribution / Novelty: Most of the methodology builds upon [Geerkens et al. (2024)] and extends it to hidden layers. The novelty therefore lies primarily in leveraging previous work within the context of XAI for intermediate layers, which remains limited. The technical contribution - formulating the current problem as an input to the prior framework - appears rather straightforward.
2. Self-containment: The paper does not meet minimal readability requirements. Key definitions are omitted, making the paper non–self-contained and difficult to understand without consulting prior work. Specifically, $QI^2$ is not formally defined (only $QI^2R$ is provided). Furthermore, formal definitions are missing for $d_{RI}$ (Eq. 4), $d_{RO}$, $T_N^2​$ (Eq. 3), and $T^2_N$​ (line 172). Every term introduced in the paper should be clearly and formally defined.
3. Writing: A "Preliminaries" section is missing, and several definitions appear in the Methodology section even though they are not newly introduced in this paper.
4. Figures: The left side of Figures 2-7 is unclear, sometimes even invisible, and lacks proper explanation on axes X,Y meaning.
5. Evaluation: A systematic evaluation with quantitative metrics is missing. The authors do not propose measurable criteria to assess their method and provide no benchmark-based analysis. Consequently, it remains uncertain whether the presented (albeit interesting) visualizations were selectively chosen.

**Questions:**

1. Extendability: The authors present their method as a general one, in contrast to other approaches (e.g., AttnLRP). To support this claim, it would be expected to include experiments on additional architectures. Why were only CNNs used for evaluation?
2. Is there any quantitative metric to evaluate the performance of the proposed method?
If the method enables network decomposition, can it also be applied to hidden-output or hidden-hidden reasoning, in addition to input-hidden reasoning? The authors are encouraged to demonstrate this capability. Otherwise, the term “network decomposition” may not be accurate.
3. In line 163, $q$ is defined as the left-hand side of the pair in Eq. (1), yet it is also described as part of $\mathcal{D}$, which consists of pairs - please clarify.
4. What do $k$ and $v$ represent in Fig. 2?
5. The left-hand images in Figures 2-7 are not sharp enough; please improve their resolution.
6. Typos: Line 272: “Figure fig. 2”.

---

### Official Review · Reviewer_LsK9 · 2025-10-31

**Soundness:** 2
**Presentation:** 3
**Contribution:** 2
**Rating:** 4
**Confidence:** 4

**Summary:**

The paper proposes a framework for providing localized, layer-level explanations of neural network behavior. It employs the QI² metric to quantify the complexity of layer responses with respect to input patches of varying sizes. The authors claim as contributions the application of QI² to neural network analysis, demonstrating the generalizability of this descriptor, the ability to analyze intermediate network abstractions, and the potential to decompose neural networks into interpretable functional substructures. The proposed framework is evaluated on two different network architectures.

**Strengths:**

- The paper proposes a framework for analyzing the **internal mechanisms of neural networks**, which can be applied across different architectures, as demonstrated with **VGG16** and **U-Net**.
- The manuscript is **well-written** and includes **illustrative examples** that showcase the approach.
- The proposed strategy has the **potential to support model inspection in safety-critical AI applications**.

**Weaknesses:**

- **Darker consistent pathways:** The concept of “darker consistent pathways” is not clearly explained. You mention that it is based on **coherence**, but it is unclear to me  **why linear relationships (lower QI² values) are considered preferable** when aggregating them across patches. A more detailed explanation or illustrative example would help clarify the rationale behind this choice.
- **Scalability and unit selection:** Analyzing the **complete network** seems impractical in this framework. How are specific units selected for analysis? Do you analyze entire layers, individual channels, or both? How an early layer impacts in the next layer’s knowledge?
- **Interpretation of features:** The claim that early layers detect **skin tone** is unclear to me. Is this determined **visually**? How many people performed the analysis to categorize the focused regions? Are there additional examples to support this claim?
- **Low- and high-level features:** The observation that features of different levels appear at various depths of the network has been described in previous xAI studies (example: Zeiler et al 2011 [1]).
- **Integration with attribution methods:** The claim that integrating QI² with methods such as **LRP** “extends traditional explainability techniques” is **debated**. It seems that QI² rather **directs the analysis** than fundamentally extends existing methods.

[1] M. D. Zeiler, G. W. Taylor, and R. Fergus, "Adaptive Deconvolutional Networks for Mid and High-Level Feature Learning," in *Proc. IEEE Int. Conf. Comput. Vis. (ICCV)*, 2011, pp. 2018–2025, doi: 10.1109/ICCV.2011.6126474.

**Questions:**

- **Segmentation experiments:** can you clarify the statement that the network exhibits a “classification-like behavior (Fig. 5a, Fig. 5b)” ?
- I believe the methodology could be **better understood if accompanied by a visual diagram**, which would help clarify the workflow and interactions between components.
- I included extra questions above.

---

> ### Comment · Reviewer_LsK9 · 2025-11-24
>
> I appreciate the clarification regarding the classification-like behavior and the interpretation of the darker, consistent pathways. Including these explanations in the final version of the paper will be very helpful for readers. I am also looking forward to seeing the additional revisions you mentioned, particularly those related to definitions and the explanation of the metric used. I recognize the value of the framework for profiling the model, so I will slightly increase my recommendation. I am not raising it further because it is difficult to assess the full extent of the planned modifications.

---

### Official Review · Reviewer_FWbX · 2025-10-31

**Soundness:** 2
**Presentation:** 2
**Contribution:** 2
**Rating:** 4
**Confidence:** 3

**Summary:**

**CRISP (Complexity-Based Reasoning of Internal Subprocessing)** is an interpretability framework that analyzes a neural network’s *internal* computations layer by layer. The method (1) decomposes activations into paired input/output patches at each module, (2) scores local transformation “complexity” using a nearest-neighbor–based QI² statistic to produce per-layer saliency profiles (MLQI²/SHLQI²), and (3) uses the most complex (salient) subregions as *anchors* to perform hidden-layer attributions (LRP-CMP) and qualitative inspection.

**Key ideas**
- **Complexity profiling:** Quantifies where a layer performs the most non-trivial transformations by comparing local neighborhoods before vs. after the layer.
- **Anchor selection:** Picks high-SHLQI² regions as representative internal “subprocesses” to analyze and attribute.
- **Hidden-layer attribution:** Propagates relevance from these anchors to visualize how evidence flows *within* the network, not only at outputs.

**Use cases**
- Demonstrated on **VGG16** (age classification) and **U-Net** (railway segmentation), yielding layer-wise narratives (e.g., early texture/edges, mid-level parts, late recombinations).

**Claims**
- Provides a modular, model-agnostic pipeline to surface functional substructures within layers.
- Offers a complementary perspective to output-centric XAI by focusing on internal transformations.

**Strengths:**

1) **Clear, modular pipeline**
   - Decomposes each layer into input/output patches → computes QI² complexity profiles (MLQI²/SHLQI²) → selects anchors → runs hidden-layer attribution (LRP-CMP).
   - Evidence: Step-by-step equations and algorithm blocks make the method replicable; consistent use across VGG16 and U-Net.

2) **Focus on internal processes (beyond output-centric XAI)**
   - Anchors are chosen inside the network (high-SHLQI² regions), enabling attribution and visualization of *internal* subprocessing rather than only final logits.
   - Evidence: Layer-wise narratives (early texture/edge processing, mid-level structure, late recombination) align with known CNN stages.

3) **Model- and task-flexible design**
   - Patchwise analysis and nearest-neighbor–based QI² scoring are architecture-agnostic in spirit and apply to different tasks (classification, segmentation).
   - Evidence: Demonstrations on VGG16 (age classification) and U-Net (railway segmentation) without changing the core pipeline.

4) **Actionable diagnostics**
   - SHLQI² profiles highlight “where” complex transformations occur, guiding targeted inspection, ablations, and potential network edits.
   - Evidence: Identified salient regions correspond to visually meaningful structures (e.g., rail boundaries, facial regions), informing qualitative assessments.

**Weaknesses:**

1) **Lack of baselines**
   - No head-to-head comparisons against established internal interpretability methods (e.g., concept discovery/ACE, Network Dissection, prototype/dictionary approaches, hidden-layer attributions).
   - Impact: Hard to judge whether CRISP offers improvements in fidelity, stability, or usefulness.

2) **Lack of quantitative results**
   - Results are largely qualitative; missing metrics for stability (across seeds/splits), localization (Pointing Game/IoU vs. parts), and causal impact (Δlogit/ΔIoU under ablation of salient regions).
   - Impact: Claims about “complexity hotspots” and functional substructures are not empirically substantiated.

3) **Insufficient human evaluation**
   - Claims are illustrated with a few images but lack systematic human studies (e.g., nameability/consistency of discovered patterns, expert ratings, task-relevance assessments).
   - Impact: No evidence that surfaced internal structures align with human concepts or aid human understanding in a repeatable way.

**Questions:**

See weakness

---

### Official Review · Reviewer_82oy · 2025-11-02

**Soundness:** 1
**Presentation:** 3
**Contribution:** 1
**Rating:** 2
**Confidence:** 4

**Summary:**

The paper proposes a method to study input-output transformations between convolutional layers, named CRISP. CRISP first decomposes a feature map from convolutional layers into patches. For each patch, CRISP computes a set of neighbors up to distance d, where each tuple is a pair of input patch and resulting patch from two layers. The framework then assigns each neighborhood a QI score to estimate the (non-)linearity within it. The collection of QI scores is visualized in a 3D histogram, which users can use to identify patterns and guide downstream tasks (e.g., attribution, layer structure analysis, etc.). The paper showcases the advantages of the framework by analyzing two deep neural networks in detail.

**Strengths:**

- The usage of QI is original and seems interesting as a general tool for patch-based work in interpretability.
- The paper is clear and easy to read, even though some details could have been included in the main text (e.g., the identification of the patch dimensions through the receptive field).

**Weaknesses:**

- **Weak experimental setup**: The main weakness of the paper is the experimental section. The paper presents two visual analyses of deep neural networks using its method. However, there are no quantitative results in terms of metrics or user studies, and no alternative methods or frameworks are compared against. Together, these weaknesses result in a lack of contextualization for the benefits of the proposed method and make the work difficult to extend. Specifically:
   - **There is no comparison against other methods**. The only attempt is in the first section, comparing a heatmap produced by the proposed method with one reported in the LRP paper. A comparison for a single image is not enough and also the models on which they are computed are not the same since authors re-trained the network. The authors state that their method should be considered complementary rather than an alternative to attribution methods. If this is the case, an experimental setup could compare scenarios where users have access only to (multiple, not just LRP) attribution methods and those where they can combine (multiple) attribution methods with the proposed method. In addition, I believe the work seems similar in principle to the literature on Deconvolution, and specifically on the feature visualization method explained in the DeconvNet paper. Contrary to the related work section, DeconvNet does not use predictions or class information and achieves a similar goal of explaining transformations of the input through the layers. The authors are encouraged to explore the DeconvNet related literature for potential competitors.
   - **There are no metrics or quantitative measurements to measure the progress**. Beyond competitors, the method is tested visually without any metric that could help users or researchers express the improvement and novelty introduced by the paper. If the authors believe that current metrics are inadequate (as stated in the paper), they should propose new metrics to show what current methods miss and how the proposed method improves on them. Metrics are also important for future research. Researchers building on this work will struggle to measure progress otherwise. If no quantitative metrics are possible, at least a user study demonstrating that the proposed method aids interpretability compared to classic frameworks would be useful.

- **Reliance on user guidance** (connected to the previous point): This is an unclear point from the current description. From my understanding, the analysis is guided by the SHLQI histogram and there is no end-to-end automation for explanations. If so, the framework is better suited as a visual analytics tool rather than a standalone framework, and would benefit from user studies focused on specific tasks. Conversely, if the framework can be considered as standalone and the histogram can be bypassed for downstream tasks (such as by selecting patches that maximize an objective connected to QI), this automation should be properly evaluated as previously explained.


- **Claims about generalization**: The paper emphasizes the property of being model-agnostic and rules out potential competitors (e.g., work in mechanistic interpretability) based on this claim. However, the paper is focused and tested solely on convolutional neural networks, explicitly relying on receptive fields to identify patches. While it may be possible in principle to identify patches in other architectures, their semantic meaning would differ substantially compared to CNNs. As stated in the paper’s limitations, *“Demonstrating this generality empirically remains important future work”*. If so, claims about being model-agnostic should be removed and alternative competitors that work in CNN should be considered. Alternatively, the framework should be tested on at least a couple of different architecture families to support generalization claims.

**Questions:**

See weaknesses for general concerns. In this section, I would like to ask authors their rationale for excluding Deconvolution and all the related work that analyze how the input is transformed through the network.

---

> ### Comment · Reviewer_82oy · 2025-11-26
>
> Thank you for the detailed rebuttal. Could the authors please upload a temporary version of the paper with the modifications and additions highlighted in a different color? This would facilitate a clearer visualization of the progress made. From my understanding, it appears that there has been progress on two of my concerns, but no progress regarding the evaluation setup. A visual highlight of the modifications would help speed up the assessment of these improvements. Thank you.

---

### Author Response · Authors · 2025-11-21
**Rebuttal [1/3]**

We would like to thank the Reviewers 82oy, FWbX, LsK9 and jTKy for their valuable feedback on our paper. Your reviews make it clear that, despite the critical points you raise, you see genuine potential in the central idea: using complexity-based, layer-wise analysis to decompose neural networks into coherent internal subprocesses and to reason about how models process information, not just what they predict. We will soon upload a revised version covering the suggestions made by you.

# Quantitative evaluation, metrics & “cherry-picked” visuals
*82oy, FWbX, jTky*

We fully agree that having quantitative criteria or human evaluation would make it easier for others to build on. Our difficulty is that almost all established XAI metrics are *output-based* (faithfulness of pixel attributions to the final prediction, infidelity, sensitivity, etc.), whereas CRISP’s focus is *function-based at the layer level: we characterize how a module transforms representations*.

This mismatch means existing metrics are not directly applicable; they would first need to be reframed in terms of (i) coherence and stability of local transformation regimes and (ii) their alignment with known internal structure. Furthermore, we currently have no baseline to compare against. To summarize, for the current submission quantitative measures would not post a benefit in understanding the approach or the presentation of the findings. However, as we agree on their importance for future work and we will explore function-level metrics and a user studies comparing “attribution only” vs. “attribution + CRISP”. If you are aware of existing metric frameworks designed specifically for function-level or intermediate-layer explanations, we would be very grateful for pointers and would be happy to integrate them into future versions.

The visualizations shown in Section 4 are *representative views of analyses performed on many inputs per module*: the same complexity hotspots and subprocess patterns reappear consistently across images, which we will state more prominently to address the “cherry-picking” concern.

# Lack of baselines / comparison to other internal interpretability methods
*82oy, FWbX*

In this first version, we chose depth over breadth: two detailed, layer-wise case studies rather than a wide sweep over concept-discovery and dictionary-style methods, which would require additional infrastructure (concept labels, unit-level annotations, extra regularization) unavailable in our current setups.

CRISP, in contrast to the mentioned methods, is concept-free and function-based: it groups patch-level input–output transformations into coherent subprocesses via QI² and then visualizes them with a generic attribution method. This objective mismatch makes standard concept/dictionary scores an imperfect proxy for how well CRISP decomposes the layer-wise function.

We will clarify this positioning in the paper:
* CRISP is complementary to concept- and prototype-based tools: it provides a structured, layer-level entry point (subprocesses) on top of which such methods or attributions can then be applied.
* In the VGG16 case study, we already combine CRISP with LRP and compare the resulting functional breakdown (Section 4.1) to the single-image explanation in Lapuschkin et al. (2017); we do **not** just compare two isolated heatmaps as the review suggests, and we will spell this out more clearly to avoid that misunderstanding

# Role of DeconvNet and feature visualization, extendability
*82oy, jTky, LsK9, FWbX*

Thank you for stressing the connection to DeconvNet and the integration with attribution methods. Our current related work section indeed undersells DeconvNet feature-visualization character. Furthermore, our wording is too strong. CRISP does **not** change the core mechanics of methods like LRP; instead, it changes the starting point and context.

Conceptually, DeconvNet, Attribution methods and CRISP answer different questions:
* **DeconvNet**: “Which input pattern activates this unit/filter?”
* **Attribution methods**: “Which input pattern are relevant to the prediction?”
* **CRISP**: “Which coherent transformation behaviors does this module implement over its input–output space?”

CRISP is population- and function-centric, treating attribution as a plug-in visualization step rather than the core explanatory mechanism. In that sense, CRISP extends the **scope of use** of the attached method. For attribution methods, it serves as guidance and starting point from purely output-dependent explanations to structured, layer-based functional analyses. For DeconvNet-related methods, it could serve as further guidance in narrowing the choice of unit and filter to explain.

---

> ### Author Response · Authors · 2025-11-21
> **Rebuttal [2/3]**
>
> # User guidance vs. automation; visual analytics vs. “standalone framework”, scalability
> *82oy, LsK9*
>
> you are correct that, in its current form, CRISP is expert-guided and might be infeasible in terms of a naïve “analyze every unit” strategy. After computing MLQI² and visualizing SHLQI², an analyst inspects the histograms, identifies coherent regions or outliers, and then explores the underlying patch transformations (optionally with LRP). We do not currently perform an end-to-end automatic selection of patches or subprocesses from SHLQI². Our intention with this work is therefore best understood as introducing a **visual analytics framework for layer-level functional analysis**, rather than a fully standalone automatic explanation pipeline. However, SHLQI² naturally lends itself to automation (e.g., automatically selecting coherent pathways or high-complexity regions under QI²-based objectives). While such an automated selection module is **not yet part of the current submission**, we see it as a promising and technically straightforward extension of our framework and will discuss this as a concrete avenue for future work in the revised version.
>
> Our current operating workflow for analysis of a network is as follows:
> 1. **Layer-level scan**: For a given architecture, we compute SHLQI² for all layers of a chosen type (e.g., convolutional layers, encoder/decoder blocks) over a representative input set. This yields a compact “complexity profile” of the network and an initial processing chain.
> 2. **Layer selection**: We identify layers with pronounced hotspots or distinctive coherent processing characteristics and focus detailed analysis on them.
> 3. **Subprocess extraction**: Within those layers, we conduct an exhaustive analysis over a multitude of inputs to strengthen hypothesis of processing.
> 4. **Linking across layers**: We then connect these subprocesses along the network to form processing chains
>
> # Generality beyond CNNs and the notion of “network decomposition”
> *jTKy, 82oy*
>
> Our claim of generality is meant in terms of *principle*, not architecture coverage in this first set of experiments: CRISP operates on local neighborhoods in activation space and only requires access to input–output patch pairs of a layer, plus a backward attribution method. None of the core components (QI², SHLQI², subprocess extraction) rely on convolutional structure or on properties specific to CNNs. In the current paper we chose convolutional classifiers and a U-Net segmentation model because they are well-established benchmarks for internal interpretability, and it is known that they somewhat exhibit structured processing that is to be visualized and explained concretely by our approach.
>
> From a practical standpoint, extending CRISP to additional architectures (e.g., Transformers, attention layers, MLPs) mainly requires engineering effort: defining suitable “patches” for their hidden representations (e.g., tokens or heads instead of spatial patches), integrating with their attribution tools, and running large-scale experiments. Given limited compute and manpower, we prioritized depth over breadth—two detailed CNN-based case studies where we could thoroughly validate the analysis—rather than adding shallow results on several architectures.
>
> Regarding “network decomposition”: in our current experiments we already analyze **hidden–hidden** mappings—each selected block is treated as a mapping from an input activation tensor to an output activation tensor of a hidden unit, and CRISP decomposes this mapping into subprocesses. QI² only requires pairs of vectors, so the same pipeline can be applied to input–hidden or hidden–output links once suitable patch decompositions are defined.
>
> # 6	“Classification-like behavior” in segmentation
> *LsK9*
>
> The phrasing is potentially confusing. We will clarify that we refer to internal behavior, not to the final task.
>
> * At the **SHLQI² level**, early encoder layers exhibit the characteristic histogram shape that QI² associates with classification tasks: steep rises in complexity and sudden drops past a threshold when neighborhoods first include points of another class. This characteristic of classification tasks is stated in the original QI² paper.
> * At the **functional level**, these layers separate sky-related pixels from the rest, consistently assigning sky patches to a distinct regime used later to suppress rail candidates in the sky region. In that sense, the encoder implements an internal “sky vs. non-sky” classification, even though the overall objective is segmentation.
>
> We will revise the text to explicitly say “classification-like internal dichotomy” to avoid suggesting that the output task becomes classification.

---

> > ### Author Response · Authors · 2025-11-21
> > **Rebuttal [3/3}**
> >
> > # "Darker consistent pathways” and why low complexity can matter
> > *LsK9*
> >
> > The term refers to a pattern in SHLQI², not to a preference for simple functions. For each neighborhood size k, SHLQI² shows the distribution of complexity values over patches, with grayscale intensity indicating the proportion of patches in a bin, normalized per column.
> >
> > A “dark pathway” is a narrow band that stays dark across multiple k: many patches are processed with similar complexity over a range of neighborhood sizes. In the example we discuss this band lies in a low-complexity region, but what makes it interesting is its **coherence and persistence**, not low QI² per se. High-complexity spikes can highlight rare idiosyncratic behaviors; dark pathways reveal simple but heavily reused subprocesses (e.g., broad filtering/masking) that are crucial to the network’s computation. We will add a short explanatory example to make this rationale clearer.
> >
> > # Self-containment and poor definition
> > We appreciate the detailed feedback and agree that the current presentation makes the paper harder to read in isolation. In the revision, we will:
> >
> > * formally define QI² and QI²R. QI² will be introduced as the general framework that maps a set of input–output pairs and a neighborhood specification to per-sample complexity values, while QI²R will be explicitly defined as the concrete score used in our experiments.
> > * explicitly define the distance terms $d_{RI}$ and $d_{RO}$ instead of relying on the generic $d(\cdot,\cdot)$ in the text. In the revised version, both will be stated as Euclidean distances in input- and output-representation space for our experiments, with a brief note that other metrics could be substituted if appropriate.
> > * clarify the definition of $T_N^2$ as the set of all pairwise combinations of patches drawn from a given tuple $T_N$, together with a short verbal explanation so that the role of this set in the QI² computation is unambiguous.
> > * clarify that $\mathbf{q}$ denotes a single patch representation that already concatenates input and output information of that patch. We will define
> > $\mathcal{D} = \\{ \mathbf{q_p} | \mathbf{p} \in \mathcal{P} \, \mathbf{q_p} =  \[ \begin{matrix} \mathbf{z_p^{l-1}} \\ \mathbf{z_p^l} \end{matrix}  \]^T \in \mathbb{R}^{d_{in} + d_{out}} \\} $
> > where $\mathbf{z}_p^{l-1}$ and $\mathbf{z}_p^l$ are the input and output patch vectors at position $p$ for layer $l$.
> >
> > We will reorganize the QI² introduction as preliminaries to ensure that every symbol is defined before use, so that the paper becomes self-contained for readers not familiar with Geerkens et al.

---

### Author Response · Authors · 2025-11-27
**Temporary highlighted version of the paper**

Thank you for the suggestion of a version with highlighted changes. We uploaded such a version to increase readability.
Additions are marked as blue and changes are marked as yellow.

---

> ### Author Response · Authors · 2025-11-27
> **Problem with figures**
>
> We just noted that the left hand sides of the figures 5 to 9 are not realy visible in the document if you open it directly from this side. However, they are visible as intended if you download the paper and then open it.

---

### Author Response · Authors · 2025-12-02
**Summary of the revisions**

Dear Area Chair,
Below, we present a short summary of the main changes and our responses which are acknowledged as detailed by reviewer 82oy and reviewer LsK9, who also raised their evaluation. For detailed information we refer to our main comment:

1. **Positioning and method clarification**
We now clearly present CRISP as a visual analytics framework that allows for human-driven layer-level functional analysis, rather than a fully automated xAI method. The workflow is explicitly described: layer-level screening with SHLQI² → selection of interesting layers → subprocess extraction → linking them into processing chains.

2. **Relation to prior work**
We clarified how CRISP differs from deep visualization methods like DeconvNet. CRISP is population- and function-centric, treating explanatory mechanism as a plug-in step rather than the core.

3. **Quantitative metrics and evaluation**
Several reviewer asked about quantitative evaluation and suitable metrics. We expanded our discussion and explicitly stated that we agree that quantitative evaluation would support future work. However, we explained that standard metrics used for xAI do not directly apply to CRISP, because CRISP provides function-level, layer-level decompositions rather than pointwise importance maps. We argue that designing appropriate function-level metrics is an open research problem and part of our ongoing work. We outline plans for user studies comparing “attribution-only” setups with “attribution + CRISP” to assess whether CRISP improves human understanding and decision-making, positioning this as the natural next step beyond the current qualitative validation.

4. **Readability and Clarification of minor points.**
We improved the readability of our paper regarding
     * Self-containment of the QI² method
     * Splitting methodology section into preliminary and method
     * Classification-like behavior at SHLQI²- and component-level
     * Generality of the framework
     * Notion of SHLQI²-specific characteristics

---

### Meta-Review · Area_Chair_RejA · 2026-01-06

**Summary:**

The primary and most consistent concern across all reviewers (82oy, FWbX, LsK9, jTky) is the significant lack of quantitative evaluation and comparison against established baselines. Reviewers argued that the submission relies almost entirely on qualitative visual inspection without standard XAI metrics (e.g., faithfulness, stability, localization) or a systematic user study to validate the utility of the proposed "Visual Analytics" framework. Secondary concerns focused on the paper not being self-contained, with key definitions missing or relying heavily on prior work, as well as poor figure visibility in the initial PDF. Reviewers also questioned the claims of generalization, noting that while the method is theoretically model-agnostic, it was only empirically tested on CNN architectures.

**Reviewer Concerns:**

The authors successfully addressed the presentation and clarity issues, specifically by providing formal definitions for the missing mathematical terms and fixing the visibility of the figures, which Reviewer LsK9 acknowledged as helpful. The authors also clarified the scope of the paper, effectively reframing it as a human-in-the-loop visual analytics framework rather than a fully automated explanation method, which helped resolve some confusion regarding scalability. However, the core deficiency regarding evaluation remains outstanding. The authors argued that existing output-based metrics are inapplicable to their function-based approach and that user studies are future work; while this explains the absence of data, it does not satisfy the reviewers' (especially 82oy and FWbX) requirement for objective evidence of the method's efficacy or superiority over existing techniques like DeconvNet or concept discovery.

**Reviewer Scores:**

Reviewer LsK9 explicitly stated in the discussion that they would "slightly increase" their recommendation following the clarifications on definitions and "darker pathways," suggesting a potential move from a score of 4 to a 6. Reviewer 82oy, who rated the paper a 2, acknowledged the clarity improvements but explicitly noted that there was "no progress regarding the evaluation setup," implying their score would likely remain at a 2 or potentially reach a 4 if they weighted the presentation fixes heavily.  Given that their primary objections (lack of metrics, baselines, and rigorous validation) were defended against rather than resolved with new data, their scores would likely remain strictly at 4 and 2, respectively.

---

### Decision · Program_Chairs · 2026-01-26

Reject